# Presenilin-1 (PSEN1) Mutations: Clinical Phenotypes beyond Alzheimer’s Disease

**DOI:** 10.3390/ijms24098417

**Published:** 2023-05-08

**Authors:** Youngsoon Yang, Eva Bagyinszky, Seong Soo A. An

**Affiliations:** 1Department of Neurology, Soonchunhyang University College of Medicine, Cheonan Hospital, Cheonan 31151, Republic of Korea; astro76@naver.com; 2Graduate School of Environment Department of Industrial and Environmental Engineering, Gachon University, Seongnam 13120, Republic of Korea; 3Department of Bionano Technology, Gachon University, Seongnam 13120, Republic of Korea

**Keywords:** presenilin-1, Alzheimer’s disease, frontotemporal dementia, motor diseases, dementia with Lewy bodies, acne inversa, dilated cardiomyopathy, risk modifier

## Abstract

Presenilin 1 (PSEN1) is a part of the gamma secretase complex with several interacting substrates, including amyloid precursor protein (APP), Notch, adhesion proteins and beta catenin. PSEN1 has been extensively studied in neurodegeneration, and more than 300 PSEN1 mutations have been discovered to date. In addition to the classical early onset Alzheimer’s disease (EOAD) phenotypes, PSEN1 mutations were discovered in several atypical AD or non-AD phenotypes, such as frontotemporal dementia (FTD), Parkinson’s disease (PD), dementia with Lewy bodies (DLB) or spastic paraparesis (SP). For example, Leu113Pro, Leu226Phe, Met233Leu and an Arg352 duplication were discovered in patients with FTD, while Pro436Gln, Arg278Gln and Pro284Leu mutations were also reported in patients with motor dysfunctions. Interestingly, PSEN1 mutations may also impact non-neurodegenerative phenotypes, including PSEN1 Pro242fs, which could cause acne inversa, while Asp333Gly was reported in a family with dilated cardiomyopathy. The phenotypic diversity suggests that PSEN1 may be responsible for atypical disease phenotypes or types of disease other than AD. Taken together, neurodegenerative diseases such as AD, PD, DLB and FTD may share several common hallmarks (cognitive and motor impairment, associated with abnormal protein aggregates). These findings suggested that PSEN1 may interact with risk modifiers, which may result in alternative disease phenotypes such as DLB or FTD phenotypes, or through less-dominant amyloid pathways. Next-generation sequencing and/or biomarker analysis may be essential in clearly differentiating the possible disease phenotypes and pathways associated with non-AD phenotypes.

## 1. Introduction

The presenilin-1 (*PSEN1*) gene was verified as one of the main causative factors for early onset Alzheimer’s disease (EOAD). The PSEN1 protein is composed of nine transmembrane (TM) domains connected with loops inside the cytosol and lumen. Two catalytic aspartates were identified in PSEN1, Asp275 and Asp385, which are located in TM6 and TM7, respectively [1]. As part of the gamma secretase complex, PSEN1 interacts with several proteins, including amyloid precursor protein (APP), Notch, nicastrin, and the modifier of cellular adhesion (MOCA), beta-catenin. Additional interacting partners of PSEN1 could be proteins or protein groups involved in apoptosis, the metabolism of calcium or cell adhesion. These findings suggest that PSEN1 could be a multi-functional protein [2,3]. First, PSENs were verified to play a role in APP cleavage and amyloid-beta generation. The majority of *PSEN1* mutations were confirmed to impact amyloid-beta production, either by reducing the production of short amyloid-beta (especially Ab40) or increasing the production of long amyloid (Ab42), resulting in an elevated Ab42/40 ratio [4]. In addition, it was suggested that PSENs also contribute to neurodegeneration through alternative pathways. For example, they play an important role in Notch signaling. PSEN1 can cleave Notch components (such as Notch1) and facilitate its transport through the Golgi system to the cell membrane [5]. Normally, PSEN1 may induce cell survival since PSEN1 mutations (such as Leu286Val or Met146Val) may result in an elevated ratio of apoptosis through amyloid-independent pathways, such as mitochondrial deficits or a calcium imbalance [3].

The discovery of *PSEN1* mutations in AD and other diseases may be a promising target in gene therapy. Currently, there is no effective therapy for the majority of neurodegenerative diseases, including genetic diseases. However, gene therapies may be promising approaches in their treatment since they have been successfully used against spinal muscular atrophy (SMA). The functional replacement of a mutant gene in children with SMA was associated with improved motor function [6]. Similar studies on treating genetic AD by “fixing” the mutant PSEN1 are ongoing. However, these studies are only tested in in vitro cell models. Human iPSC lines from an AD patient with Leu150Pro and Ala79Val were also targeted to CRISPR-Cas9 gene correction, and both approaches were successful. These studies may be useful in future studies of gene therapy [7,8]. CRISPR-Cas9-based treatment was used against PSEN1 Met146Leu in fibroblasts. Guide RNAs (gRNAs) were designed against mutant PSEN1. Knocking out the mutant PSEN1 was associated with a reduction in the Aβ42/40 ratio [9]. Gene silencing may also be promising in cases of PSEN1 mutations. HeLa cells and fibroblasts with PSEN1 Leu392Val mutations were targeted to antisense siRNAs, which were designed for the mutation (with one mismatch to the normal gene). The gene silencing reduced the toxic Aβ42 production [10]. Identifying the individual disease risk (including for a PSEN1 mutation carrier) at the pre-clinical stage should be essential for future therapy, since gene therapies may be effective before clinical symptoms appear [9].

To date, more than 300 mutations have been reported in PSEN1. Phenotypic heterogeneity was associated with mutations in PSEN1 (https://www.alzforum.org/mutations/psen-1, accessed on 1 September 2022). The age of onset may be variable; several cases were related to a young disease onset (late twenties or early thirties), such as Leu85Pro [11], His163Pro [12] or Met233Val [13]. Meanwhile, other mutations, such as Met84Val [14] Ala164Val [15] or Phe175Ser [16], were associated with late disease onset (late sixties and seventies). In addition to the EOAD phenotypes, several different symptoms were related to PSEN1 mutations, including myoclonus, seizures, extrapyramidal symptoms, spastic paraparesis and behavioral and language dysfunctions [17,18,19]. Several cases with PSEN1 mutations did not fulfill the AD diagnostic criteria (CERAD and NIA-Reagan criteria) but presented other disease phenotypes, such as frontotemporal dementia (FTD), progressive non-fluent aphasia or Pick’s disease and diseases with motor impairment, such as Parkinson’s disease (PD) and dementia with Lewy bodies (DLB). Additionally, two PSEN1 mutations presented non-neurodegenerative phenotypes, including acne inversa or dilated cardiomyopathy (Table 1) [17,18,19].

This manuscript will introduce the PSEN1 mutations described in disease phenotypes other than EOAD. We will also discuss the potential mechanisms in which PSEN1 could be involved in other forms of neurodegenerative diseases, including frontotemporal dementia or motor dysfunctions. This review will also introduce how PSEN1 could impact the onset of non-neurodegenerative disorders, including acne inversa and cardiomyopathy.

## 2. Potential Impact of PSEN1 on Other Forms of Neurodegenerative Disease

### 2.1. PSEN1 and Frontotemporal Dysfunctions (FTD; Pick’s Disease)

FTD or AD with FTD-like symptoms (behavioral or language variant disease) were observed in several PSEN1 mutations, since at least one case with Leu113Pro, Gly183Val, Leu226Phe Met233Leu or Arginine insertion in residue 352 presented FTD -like symptoms (Table 2). AD with Pick’s bodies appeared in patients with Met146Leu or Ala260Val mutations. Progressive non-fluent aphasia may be present in PSEN1 mutations, including Pro264Leu. AD and FTD were verified to share similarities in clinical phenotypes and symptoms. Several AD patients present frontotemporal brain damage with behavioral or language dysfunctions. In addition, FTD could be associated with severe cognitive decline in patients. A differential diagnosis of AD and FTD may be challenging [20]. 

In the case of PSEN1 Leu113Pro, mutation-carrying family members fulfilled the requirements of the Lund and Manchester criteria [21,22] and consensus criteria [23] for FTD. Imaging (CT and SPECT) revealed neurodegeneration in the frontal lobes. However, the neuropathology of the patients presented typical AD pathology with senile plaques and tangles, suggesting that the patients had frontal variant AD rather than FTD [24]. A post-mortem analysis in an FTD patient with Met146Val revealed both FTD (Pick’s disease)- and AD-related neuropathological hallmarks. Senile plaques and tangles were prominent in the brain, but Pick’s bodies were also detected in the hippocampus and cortex. In addition, ubiquitin-positive nuclear inclusions appeared in the white matter oligodendrocytes. Frontal, temporal, and hippocampal atrophy were observed in the brain, with enlarged ventricles. Depigmentation was also found in the substantial nigra [25]. However, other cases with Met146Val presented typical AD-related pathology [26]. PSEN1 Gly183Val was associated with Pick’s disease and met the consensus criteria for FTD. Post-mortem studies revealed FTD-related changes. The brain presented atrophy in the frontotemporal region, and in the neocortex, Pick’s bodies and Tau-positive inclusions appeared, but amyloid plaques were missing [27]. However, the second case of mutation (the co-existence of Gly183Val and Pro49Leu) presented only mild atrophy in the frontotemporal gyri and typical AD neuropathology without Pick’s bodies [28]. One patient with Leu226Phe was initially diagnosed with FTD (Lund and Manchester consensus), but a post-mortem analysis (CERAD and NIA-Reagan criteria) suggested that the patient may have presented with frontal variant AD [29]. A patient with an Arg352 duplication presented FTD and Parkinsonism with ubiquitin positive inclusions. However, the pathogenic nature of the mutation may be questioned since the patient also carried a splice-site mutation in the progranulin (GRN) gene [30]. Patients with PSEN1 Met233Leu and Val412Ile also fulfilled the clinical criteria for FTD, and imaging showed frontal and parietal involvement; however, no neuropathological data were available on them [31,32].

**Table 2 ijms-24-08417-t002:** Examples of PSEN1 mutation cases which were initially diagnosed as FTD.

Mutation	Initial Diagnosis	Diagnostic Criteria	Imaging	Neuropathology	Revised Diagnosis	Reference
Glu15His	Early onset FTD	NA	NA	NA	NA	[33]
Asp40del	bvFTD	Clinical symptoms; imaging	MRI: atrophy involving the lateral frontal lobes	Amyloid deposits in frontal lobes	Frontal variant AD	[34]
Leu113Pro	FTD	Lund and Manchester criteria	CT: frontotemporal atrophy; SPECT: hypoperfusion in frontal lobes	AD-like plaques and tangles	May be frontal variant AD	[24,35]
Thr122Ala	FTD	NA	NA	NA	NA	[33]
Ala137Thr						
Met146Val	bvFTD	Lund and Manchester criteria	Frontal temporal atrophy	Co-existence of amyloid plaques, tangles, and Pick’s bodies	AD and Pick’s disease co-existence	[25,26]
Gly183Val	FTD	Clinical symptoms, imaging	Severe frontotemporal atrophy	Neocortex: Pick bodies, Tau-positive cytoplasmic neuronal inclusions, and no amyloid plaques	Pick’s disease	[27]
Leu226Phe	FTD	Lund and Manchester criteria	frontal atrophy, no hippocampal atrophy, and hypoperfusion in frontal area	Typical AD hallmarks	Frontal variant AD	[29]
Met233Leu	FTD	Clinical Consensus Criteria for FTD	PET: hypometabolism in prefrontal, parietal, and temporal cortices	NA	FTD	[31]
Pro303Leu	FTD	NA	NA	NA	NA	[33]
Arg352dup	FTD	Clinical Consensus Criteria for FTD	Fronto-temporoparietal cortical and right hippocampal atrophy	Ubiquitin-immunoreactive structures	FTD, also carried a GRN splice site mutation	[30]
Pro355Ser	FTD/frontal variant AD	Clinical symptoms, imaging	MRI: microbleeds in cortex; FDG-PET: frontotemporal hypometabolism	Suspected Lewy bodies	Frontal variant AD	[36]
Val412Ile	FTD	Clinical symptoms, imaging	FDG-PET: hypometabolism in the parietal and frontal cortices	NA	FTD	[32]

The impact of PSEN1 gene in FTD phenotypes remains unclear; however, the role of PSEN1 in FTD may not be ruled out. PSEN1 could possibly contribute to FTD through loss-of function mechanisms or amyloid-independent mechanisms. Loss-of function mutations result in reduced gamma secretase mechanisms and amyloid cleavage, which could also lead to a neurodegenerative process through different mechanisms, including Tau phosphorylation, protein trafficking, apoptosis, and calcium balance [37]. A study by Amtul et al. analyzed the duplication mutation in Arg352, which was observed in patients with familial FTD. This mutation was found to inhibit the presenilin-mediated gamma secretase activity. The lower gamma secretase activity could reduce the cleavage of the toxic C-terminal fragment (CTF) of APP, which may accumulate in the brain. In addition, mutation may reduce the gamma secretase activity towards other substrates, such as Notch. This study did not provide an exact answer as to how the mutation may contribute to FTD; it may be possible that the chronic inhibition of the gamma secretase cleavage of an unknown substrate may lead to FTD-related neurodegeneration [30,38]. 

PSEN1 Gly183Val was confirmed to impair the splicing of the PSEN1 transcript. The aberrant splicing due to PSEN1 Gly183Val could result in the absence of exon 6 or 6 and 7 from the PSEN1 transcript and a premature STOP codon in exon 5. This may result in loss-of function mechanisms, such as reduced PSEN1 expression [27]. A mouse study on PSEN1 Gly183Val mutation revealed lower full-length PSEN1 mRNA expression in the mouse brain, leading to a reduction in PSEN1 function [39]. The Leu113Pro mutation was also found in patients diagnosed with familial FTD [35]. Leu113 was verified as a critical residue in gamma secretase processing since it impacts the cleavage of amyloid intermediates to shorter forms. Leu113Pro mutation may play reduce this mechanism, resulting in reduced total amyloid levels [40]. Leu113 is located at a splicing site, leading to aberrant PSEN1 transcripts [35]. A study by Evin et al. (2002) analyzed PSEN1 in several neurodegenerative diseases, which could also be proof of the role of abnormal PSEN1 splicing in FTD. The alternative PSEN1 transcript (42 and 46 kDa) was either prominent or detectable in brain samples from FTD patients. An RT-PCR analysis suggested that the splicing occurred inside exon 8 [41]. Future studies are needed to determine how the abnormal splicing of PSEN1 could contribute to FTD or FTD-like phenotypes [39]. 

Another putative mechanism which may relate PSEN1 mutations to FTD could be related to impaired Tau-related pathways. PSEN1 can potentially impact Tau phosphorylation [31]. Shepherd et al. (2004) revealed that plaque-associated Tau deposition may be prominent in cases with PSEN1 mutations compared to the sporadic AD cases. Post-mortem studies revealed that different PSEN1 mutations, such as Met146Leu or Pro264Leu, were associated with Tau-positive plaques. Mutations may increase the insoluble Tau levels and potentially impact the phosphorylation of the Tau protein [42]. Experiments on mouse models with PSEN1 suggested that mutation may impact Tau phosphorylation but not the NFT formation. PSEN1 and Tau interaction may play a key role in the onset of FTD [43,44]. PSEN1 would normally stimulate the PI3K/Akt signaling, which could play a role in cell survival and the prevention of Tau phosphorylation. Mouse experiments revealed that PSEN1 knockout (and loss-of function mutations in PSEN1) could inhibit the Akt phosphorylation and increase the degree of apoptosis. Restoring PSEN1 expression could rescue the cells from apoptosis. PSEN1 may control PI3K-Akt signaling through cadherin-dependent signals. This pathway may also impact Tau phosphorylation and p-Tau aggregation. PS1 may control Tau phosphorylation in the brain by inducing cadherin/PI3K/Akt/GSK-3 signaling. PSEN1 knockout mice showed a lower cadherin/PI3K association, resulting in reduced PI3K/Akt activity and enhanced Tau phosphorylation [45,46]. 

No evidence was found as to whether PSEN1 and progranulin (PGRN) can interact. However, the brain tissue of a patient with an Ala246Glu mutation suggested that a PSEN1-PGRN interaction may be possible. The mutation may be related to frontal variant AD without Pick’s bodies. An autopsy analysis of a pre-symptomatic carrier found granulin-positive plaques beside the amyloid plaques, which were prominent in the frontal and temporal region of the brain. PGRN was suggested to have anti-inflammatory effects and is expressed in nerve cells and neurons. A link between AD and inflammation has been verified. However, it remains unclear whether the PSEN1 mutation impacts the abnormal progranulin aggregation or whether they may be generated through a PSEN-independent mechanism [47]. Taken together, PSEN1’s involvement in the formation of frontotemporal dementia/Pick’s bodies remains unclear. However, cell and mouse models revealed that they may be related to loss-of function mechanisms, splicing or Tau phosphorylation (Figure 1). 

### 2.2. PSEN1 and Motor Impairment

Several PSEN1 mutations were associated with motor impairment, including spastic paraplegia, Parkinsonism or myoclonus. Spastic paraplegia (SP) was a quite common phenotype and is related to EOAD with PSEN1 mutations, for example Tyr154Asn, Phe237Ile, Pro264Leu and Glu280Gly, or mutations related to the abnormal splicing of exon 9 (Table 3). Amyloid-positive cotton wool plaques may be common hallmarks among PSEN1 mutation cases with spastic paraplegia (or Parkinsonism), such as the deletion of residue 84–84, Pro436Gln, Arg278Gln and Pro284Leu or the deletion of PSEN1 exon 9. However, the mechanisms of how PSEN1 may be related to cotton wool plaque formation remain unclear [48]. NFTs and cerebral amyloid angiopathy (CAA) may also be present in several patients with PSEN1-related SPs. It may be possible that PSEN1 shares common pathways with genes involved in hereditary spastic paraplegias (HSPs). A pathway analysis of PSEN1 and HSP genes revealed that the knockdown of PSEN1 may disturb neurotrophin signaling and Wnt signaling pathways. Both of these pathways may play a role in several neuronal functions, such as neuronal development, morphogenesis or survival. The impairment of Wnt- and neurtophin pathways could also disturb the development of motor neurons. Additionally, both of these pathways could be involved in AD pathogenesis via impaired neuroprotection and neuronal morphogenesis [49].

Parkinsonism was also common hallmark of EOAD with PSEN1 mutations. In the majority of PSEN1 mutations, the initial symptoms may be cognitive decline, and Parkinsonism occurs in later disease stages. However, Parkinsonism could also be an initial symptom in cases of several PSEN1 mutations, mutations such as Arg41Ser, Gly217Asp and Val272Ala [50,51,52]. A patient with PSEN1 Arg41Ser developed a tremor and bradykinesia and was diagnosed with PD. Treatment with L-dopa improved the clinical symptoms. FDG-PET revealed that amyloid deposition was missing, and the pattern of hypometabolism was more similar to a normal aging brain. CSF amyloid and Tau levels were normal [50]. Gly217Asp was also associated with dementia and Parkinsonism. Neuropathology revealed cotton wool plaques, which contained amyloid-beta species, fibrils and neuropil elements in several brain regions such as the cortex, caudate nucleus, putamen, claustrum, thalamus and substantia innominate. It may be possible that the cotton wool plaques could impact the onset of Parkinsonism [51]. The mechanism of how PSEN1 may be involved in Parkinsonism remain unclear. It may be possible that amyloid deposition may impact dopamine transport, particularly the presynaptic dopaminergic pathway [52].

AD and PD may share several common pathways, such as abnormally folded protein clearance, neuroinflammation, endosomal dysfunction, neuroinflammatory process, lysosomal functions, etc. It was suggested that PSEN1 interacts with PD-related genes, including PRKN (or PARK2) and PINK1. PRKN could upregulate the expression of PSEN1. As part of the gamma-secretase complex, PSEN1 could impact APP cleavage, thereby controlling the formation of the APP intracellular domain (AICD). AICD could interact with FOXO3, which enhances the Pink1 expression. AICD may impact the expression of several genes involved in mitochondrial dynamics, for example, by reducing the expression of DNM1L/Drp1 and MFN2 (mitofusin 2). In addition, AICD is involved in the expression of mitophagy/autophagy markers. AICD enhances LC3-II expression but decreases the expression of SQSTM1, TIMM and TOMM. Through these genes, upregulated PINK1 could stimulate PRKN expression and mitochondrial functions. The PRKN-PSEN1-PINK1 cascade through AICD interactions could control the mitochondrial pathways (biogenesis, organelle trafficking and mitophagy) and autophagy. PSEN1 mutations could possibly result in PD or PD-like phenotypes via the impairment of PRKN-PINK1-dependent mitochondrial processes [50,53]. PD and mitochondria have been verified to be closely related, and abnormal mitochondrial pathways could play a key role in disease progression [53].

Patients with PSEN1 mutations may present DLB (such as Ala79Val [54]), Ser132Asn [54] Gly184Asp [55] or DLB-like phenotypes (such as Thr440del [56]). In addition, several PSEN1 patients diagnosed with EOAD had also Lewy bodies in their brain, including Ser170Phe [57], Leu174Arg [58] or Met233Val [59]. Post mortem studies on a DLB patient with Ser132Asn showed AD pathology with Lewy bodies in the neocortex [60]. One patient from Japan with Glu184Asp developed DLB (with primary progressive aphasia), and neuropathology studies revealed strong alpha synuclein pathology. In addition to the Lewy bodies, non-amyloid comsponents (NACs) were accumulated in the neuritic plaques and astrocytes [56]. A second patient from the Czech Republic with Glu184Asp was diagnosed with primary progressive aphasia, but Lewy body pathology was also observed in neuropathology studies. The diagnosis for this patient was revised to EOAD with Lewy bodies [61]. 

The Japanese patient with Thr440del was also diagnosed with EOAD and Lewy bodies; however, in the brain, both Lewy bodies and cotton wool plaques were present. Neural loss also affected the substantia nigra. Due to the consensus on the DLB diagnosis guidelines, this patient was suggested to have Parkinson’s disease with dementia [57,62]. Normal PSEN1 was suggested to impact the alpha-synuclein and amyloid beta interactions or inter-functions. Studies on mutations with DLB or EOAD with Lewy bodies were suggested to induce not only the amyloid beta pathology but also the alpha synuclein pathology [57]. 

Studies on PSEN1 Leu166Pro and exon9 deletion revealed that PSEN1 may interact with alpha synuclein. This interaction may occur inside the different membrane compartments such as synaptic vesicles, Golgi apparatus or mitochondria. Mutant PSEN1 and alpha synuclein may prevent the release of the appropriate transport of alpha synuclein to phagosomes and autophagosomes. PSEN1 mutations may result in a stronger interaction between PSEN1 and alpha synuclein. This interaction may inhibit the release of alpha synuclein to proteosomes or autophagosomes, leading to the aggregation of alpha synuclein. Further studies are needed to determine how a mutant PSEN1–alpha synuclein interaction may impact the formation of Lewy bodies [63]. Taken together, PSEN1 may interact with several genes or proteins. The impairment of these gene interactions could result in motor impairment in cases of PSEN1 mutations. Figure 2 introduces the possible pathways through which PSEN1 could impact motor impairment diseases. 

**Table 3 ijms-24-08417-t003:** Examples of mutations involved in Parkinsonism or DLB as initial symptoms.

Mutation	Initial Diagnosis	Diagnostic Criteria	Imaging	Neuropathology	Revised Diagnosis	Reference
Arg41Ser	l-dopa-responsive early onset Parkinsonism	Imaging and CSF markers	MRI: moderate frontal atrophy; FDG-PET: no amyloid deposits	NA, mild increase in CSF Tau but not amyloid. No amyloid deposits	Early onset Parkinsonism	[50]
Gly217Asp	Parkinsonism, dementia	Imaging post-mortem studies	MRI, CT: atrophy of frontal and temporal regions and mild atrophy in cerebellum	Cotton wool plaques, extracellular amyloid fibrils, and tau-immunopositively neurites	Presenile dementia and Parkinsonism	[51]
Val272Ala	Parkinsonism, dementia	CERARD, Newcastle	MRI, PET: subcortical-frontal area	Typical AD and Lewy bodies	Subcortical dementia	[52]
Glu184Asp	DLB	CERARD	MRI: diffuse cortical atrophy; SPECT: hypoperfusion	Lewy bodies and no amyloid components in astrocytes	AD with Lewy bodies	[56,61]
Gly417Ala	Parkinsonism, dementia	NINCDS-ADRDA	MRI: atrophy of parietal and anterior temporal regions and cortex	PiB-PET: diffuse amyloid deposition in several brain areas (cerebellum, frontal, parietal and temporal cortices)	EOAD with Parkinsonism	[64]
Gly417Ser	Parkinsonism, AD	Imaging, post-mortem	SPECT: Hypoperfusion in different brain areas	Cotton wool plaques and Lewy bodies	AD with cotton wool plaques	[65]
Thr440del	L-DOPA responsive Parkinsonism	Consensus guidelines for DLB	Atrophy of frontal/temporal lobes and brainstem	Neural loss in substania nigra, Lewy bodies, and cotton wool plaques	Variant AD with Lewy bodies	[57]

### 2.3. PSEN1 Involvement in Non-Neurodegenerative Disease Phenotypes

One PSEN1 mutation, Pro242fs was linked to hidradenitis suppurativa (HS) or acne inversa, which is a skin disease. This mutation could result in the premature truncation of the PSEN1 protein and a lack of the C-terminal region after the fifth transmembrane domain. Acne inversa is a chronic inflammatory disease associated with skin deformities, such as acne-like lesions, follicular occlusion or progressive scarring [66]. The role of gamma secretase in acne inversa has been investigated, including PSEN1, presenilin enhancer 2 (PEN2) and nicastrin (NCT). Reduced expressions of mutant PSEN1 and NCT appeared in acne inversa patients. Since the mutations were either frameshift or nonsense variants, the reason for the reduced expression was nonsense-mediated mRNA decay [67]. 

Gamma secretase defects may result in abnormal hair follicle development and the formation of epidermal cysts. The development of sebaceous glands may also be disturbed in the case of gamma secretase deficiency [68]. The overexpression of the Pro242fs mutation in zebrafish embryos suggested that it may not increase APP processing and impact gamma secretase activity. However, the mutation was found to enhance Notch signaling [67]. Notch signaling has been recognized as a main regulating factor for cell homeostasis and was suggested to be involved in hair follicle development, the hair cycle and hair follicle differentiation. Furthermore, Notch signaling could also play a role controlling resuscitation in the case of injured hair organs [69].

It was suggested that the mutation alters the inflammatory pathway and changes the cytokine/chemokine expression in PMA-differentiated macrophages. A truncated PSEN1 protein was related to a higher expression of pro-inflammatory TNF alpha, TNFSF11 or IL11. Additional chemokines, such as CCL17 and CX3CL1, were also upregulated in mutant macrophages [66]. Furthermore, Pro242fs mutation may reduce the expression of other inflammatory molecules, such as LIF, IL5 IL12B, CSF2 and BMP2. The overexpression of the Pro242fs mutation could induce inflammation and impair the immune cell homeostasis. The mutation could result in disturbances in the differentiation of granulocytes and macrophages, the maintenance of immune reaction or the regeneration of hair follicles [66,70] (See Figure 3).

Dilated cardiomyopathy (DCM) was related to PSEN1 Asp333Gly mutation. The mutation was found in an African family in which one family member had AD, while the other six affected relatives were diagnosed with an aggressive heart disorder [71,72]. Both APP and PSEN genes were found to be expressed in the heart, suggesting that the role of APP-related gamma secretase activity in heart disease may not be ruled out. To date, no evidence has been found of the relationship between APP cleavage and heart dysfunction [73,74]. PSEN1 D333G was transfected into HEK293 cells but did not result in an alteration in the Ab42/Ab40 ratio. DCM may not be related to impaired APP cleavage by gamma secretase, suggesting that PSEN1 (and PSEN2) could impact heart failure through an amyloid-independent mechanism [72]. PSEN1 may play a crucial role in cardiac development. PSEN1 knockout mice presented several heart dysfunctions, such as ventricular septal defect (VSD) or double outlet right ventricle (DORV). PSEN1 deficiency could result in impairment in the development of cardiac neural crest cells and heart morphogenesis [75]. Possible mechanisms through which PSEN1 may be related to DCM include impaired Notch signaling [71,75,76]. Notch signaling was confirmed to be involved in heart development; as an endocardial-derived signal, it can regulate the cell fate determination and heart tissue development and regeneration. Notch signaling could interact with several molecules, such as Tgfβ2, Bmp10, Fgf8 and Bmp4, for cardiac development and function. Dysfunctions in Notch signaling may result in heart dysfunctions in both children and adults [76].

Other putative mechanisms of PSEN1 and DCM could be due abnormal beta-catenin pathways or impaired calcium signaling. Skin fibroblast studies revealed that mutation could impair the calcium balance. Mutation carrier fibroblasts presented higher intracellular calcium concertation compared to the non-carriers [71]. PSEN2 knockout mice also presented elevated cardiac contractibility and calcium transients compared to controls [77].

A PSEN1 null mutation in adult mice resulted in reduced muscle fibers, a shorter length of sarcomeres in cardiomyocytes and widened Z-lines in sarcomeres. Furthermore, the levels of beta-catenin and cyclin downstream D1 (Cycd1) were down-regulated in the hearts of PS1-KO mice. This study suggested that PSEN1-KO may be related to an impairment in cardiomyocyte relaxation, result in a lower ventricular volume and cardiac hypertrophy [78]. The cardiovascular deletion of PSEN1 in mice may also impair calcium homeostasis. The PSEN1 knockout resulted in the abnormal expression of genes involved in cardiac muscle development and calcium ion channel function. In addition, in PSEN1-KO cardiomyocytes, calcium ions were increased and decreased in the cytosol and sarcoplasmic reticulum (SR), respectively. The SR calcium was also decreased in WT mice with heart dysfunction, but its level was lower in the PSEN1-KO mice. These data reveal that PSEN1 dysfunctions may impact the release of calcium from the SR to the cytoplasm. PSEN1 knockout mice with heart failure presented lower calcium levels in the SR when compared to the wild-type mice with heart failure. PSEN1 abnormalities may result in dysfunctions of the calcium balance in failing hearts, which might impact the onset of aging-related cardiomyopathy [79] (See Figure 4). 

No association was found between PSEN1 mutations in cancer, but PSEN2 Arg62His and Arg71Trp may impact breast cancer [80]. Kim and Jeong (2020) analyzed somatic mutations in dementia-related genes in different forms of cancers. Somatic PSEN1 mutations (such as Ala246Val, Glu71Lys and Arg269Cys) appeared in six major forms of cancer, including breast carcinoma, non-small cell lung cancer, colorectal adenocarcinoma, glioblastoma, endometrial carcinoma and ovarian tumors. Further studies were found as to whether these mutations could be related to cancer progression [81]. The role of PSEN1 mutations in cancer may not be ruled out. PSEN1 expression was found to be inhibited by p53, suggesting that PSEN1 may be an anti-apoptotic protein. PSEN1 was found to impact cell survival through both p53-dependent and p53-independent pathways [82,83]. 

Studies in glioma cells revealed that PSEN1 may protect against cell proliferation through inhibiting Wnt/β-catenin signaling. The down-regulation of PSEN1 could enhance the growth of glioblastoma cells, while the overexpression of PSEN1 may prevent or slow down the beta-catenin-dependent cell proliferation of glioblastoma [84]. PSEN1 could be protective against cancer through epigenetic mechanisms. PSEN1 was suggested to be a target for the tumor suppressor microRNA-193a-3p, and their interaction may be beneficial in the case of multi-chemoresistant bladder cancer. PSEN1, controlled by miR-193a-3p, may activate the pathways related to DNA damage response [85]. Additionally, miRNA-193a-targeted PSEN1 may protect against cell proliferation and invasion and inhibit the growth of gastric cancer cells. Furthermore, miR-193a/PSEN1 could activate the PI3K/Akt pathway and prevent cancer cell differentiation from the epithelial to the mesenchymal stage [86].

## 3. Atypical Inclusions in the Brain of Patients with PSEN1 Mutations

The two main hallmarks of AD are the senile plaques and neurofibrillary tangles. However, additional atypical aggregates may also be present in case of patients with PSEN1 mutations, such as Pick’s bodies, Lewy bodies, TDP43 aggregates or gliosis (Table 4) [19]. Patients with PSEN1 mutations and FTD-like phenotypes may be associated with abnormalities in the frontal brain region (for example, the frontal-anterior temporal cortex, frontal-subcortical circuits or temporal lobe). These patients can present either amyloid plaques and neurofibrillary tangles (NFTs) in the frontal area or Pick’s bodies, but the AD pathology and Pick’s bodies may also be co-existed [27,31]. Pick’s bodies were present in several cases with PSEN1 mutations. Pick’s bodies are argyrophilic, round inclusions outside of nerve cells which may affect several brain areas, including frontal and temporal areas. However, they may be present in other brain areas as well, including the hippocampus. They are the main hallmarks of FTD, but they may occur in other diseases, such as AD [87,88]. Pick’s bodies were found in at least one carrier of PSEN1 Met146Leu [19], Met146Val [89]. Gly183Val [27], Ala263Val [90] and exon9 deletion [89]. The majority of these cases also presented with typical AD pathology; however, the patient with Gly183Val mutation did not have amyloid aggregates [27]. The patients with PSEN1 mutations and Pick’s pathology presented FTD-like symptoms [19]. One unique preclinical AD case (the individual died before exhibiting clinical symptoms) with PSEN1 Ala246Glu was positive for plaque-like PGRN aggregates. Although amyloid aggregates also appeared, the PGRN plaques were more prominent in the medial and frontal brain areas. Amyloid plaques were more dominant in the anterior cingulate gyrus. However, this is currently the only case of PSEN1 mutation with PGRN plaques, and it remains unclear whether these aggregates could play a role in an AD-related process [47]. 

Transactive response DNA-binding protein of 43 kDa (TDP-43) aggregates are hallmarks of FTD or amyotrophic lateral sclerosis (ALS). TDP43 is a ribonucleoprotein that plays a significant role in RNA regulation, including transcription, splicing and stabilization. TDP43 hyperphosphorylation, ubiquitinoylation or abnormal cleavage may result in cytoplasmatic aggregation of the protein. TDP43 aggregates may also be observed in AD patients. TDP43 plaques may co-localize with typical AD hallmarks (senile plaques and NTFs), and there may be a putative interaction between them. Additionally, TDP43 aggregates may increase the severity of AD pathology. The TDP43 aggregates in AD patients may be associated with motor impairment (limbic predominant subtype of AD). These aggregates may also increase the degree of atrophy in the hippocampus [91,92]. TDP43 pathology co-existed with alpha-synuclein and typical AD pathology in one of the patients with PSEN1 Gly206Arg and Arg278Ile mutations [93,94]. A patient with Gly417Ser had also TDP43 inclusions in the limbic region and temporal cortex; however, other inclusions also appeared in the patient, including Lewy bodies and cotton wool plaques [65]. Ubiquitin positive plaques were observed in one case of PSEN1 Glu280Ala mutation in the cerebellar areas, which were surrounded by reactive astrocytes and dystrophic neurites [95].

Abnormal alpha synuclein aggregates or synucleopathies inside neurons are common hallmarks in neurodegenerative diseases, including dementia with Lewy bodies (DLB), Parkinson’s disease (PD) or multiple system atrophy (MSA), but they were also observed in AD patients. [96,97]. The misfolded alpha synuclein could accumulate in the Lewy bodies through an unknown mechanism. Two types of Lewy bodies could be distinguished. The classical Lewy bodies could be found in the brainstem. They may be spherical and consist of an eosinophilic core and a pale halo. Meanwhile, in the cortical type, Lewy bodies may not have the halo, and they do have a less compact appearance compared to classical ones. [98,99]. AD patients presented Lewy bodies in the amygdala, periamygdaloid cortex, entorhinal cortex, brainstem or neocortex. Lewy bodies and an AD-type pathology may also co-exist [100]. Several patients with PSEN1 mutations, such as Leu170Phe [101], Leu202Phe [102] or Ala396Thr [103], were associated with EOAD and Lewy body pathologies [19]. A case with Gly184Asp was diagnosed with DLB and primary progressive aphasia [56], while another patient with a Val272Ala mutation was diagnosed with Parkinsonism and subcortical dementia [104]. 

**Table 4 ijms-24-08417-t004:** Atypical inclusions in patients with PSEN1 mutations. Abbreviations: PB: Pick’s bodies, TDP43: Transactive response DNA-binding protein of 43 kDa (TDP-43); LB: Lewy bodies, CWP: cotton wool plaques.

Inclusions	Mutation	Amyloid Plaques	Tau	Amyloid Angiopathy	Other Neuropathology	Disease	Reference
PBs	Met146Leu	Cortex	Cortex	NA	Pick bodies in upper frontotemporal cortex and dentate gyrus; ballooned neurons. Lewy bodies in amygdala.	AD with Pick pathology	[89]
	Met146Val	Frontal, parietal cortex	Several brain areas	NA	Co-existence of amyloid plaques, tangles, and Pick’s bodies; cotton wool plaques (CWPs) may also appear. TDP43-positive inclusions also appeared and Lewy bodies were also observed.	AD with Pick pathology	[25,89]
	Gly183Val	NA	Neocortex	NA	Neocortex: Pick bodies and Tau-positive cytoplasmic neuronal inclusions.	Pick’s disease	[27]
	Ala260Val	Plaque ring around blood vessels; neocortex	Neocortex	NA	Gliosis in cerebral neocortex; Pick-like intraneuronal inclusions in the dentate gyrus.	AD with Pick pathology	[90]
	Exon9 deletion	Cortex	Cortex	NA	One case of Pick’s bodies in hippocampus.	AD with Pick pathology	[89]
TDP43	Gly206Arg	Cortex, striatum and thalamus; cerebellum	Medulla oblongata, dorsal vagal nucleus, substania nigra	Meningeal vasculature	End-stage TDP-43 and alpha synuclein pathology.	EOAD	[93]
	Arg278Ile	Cortex	NA	Cortex and leptomeninges	Alpha-synuclein and TDP43 pathology in the amygdala.	EOAD	[94]
	Gly417Ser	frontal, parietal, and temporal cortices., cerebellum, spinal grey matter	NA	Parietal and anterior temporal regions	CWPs (cortex); Lewy bodies in neocortex; TDP-43 inclusions in the limbic region and temporal cortex.	Spastic Paraparesis, EOAD	[65]
Granulin aggregate	Ala246Glu	Cingulate gyrus, temporal lobe	NA	NA	PGRN aggregates in medial temporal and frontal areas. May co-aggregate with amyloid Gliosis.	Preclinical case	[47]
Ubiquitin positive plaques	Glu280Ala	Cerebral cortex, hippocampus, cerebellum, midbrain and basal ganglia		Several brain areas	Ubiquitin–positive plaques surrounded by reactive astrocytes and dystrophic neurites in the cerebellum.	EOAD	[95]
LBs	Pro117Ser	Hippocampus, temporal cortex	cortex	Intracortical vessels	Lewy bodies in the cortex in most patients.	EOAD	[105]
	Ser132Ala	Hippocampus	Hippocampus	NA	Lewy bodies in neocortex and substania nigra. Neuritic plaques in temporal cortex.	EOAD/DLB	[55]
	Ser170Phe	Hippocampus	hippocampus	Several brain areas	Ubiquitin-positive oval intraneuronal inclusions and neuritic plaques in dentate fascia. Lewy bodies in brainstem, limbic areas, and neocortex.	EOAD with Lewy bodies	[101]
	Leu174Arg	Isocortex	Allo- and isocortex	NA	Lewy bodies and amygdala and entorhinal cortex.	EOAD	[59]
	Glu184Asp	Cortex, cerebellum	Cerebral cortex, amygdala, thalamus, substantia nigra	Leptomeningeal and parenchymal arteries in cortex, brainstem, cerebellum	Lewy bodies and non-amyloid components in plaques and astrocytes.	DLB + primary progressive aphasia	[56]
	Leu202Phe	Hippocampus and several brain areas	Several brain areas	Cortical and leptomeningeal blood vessels	Lewy pathology in the amygdala. Gliosis in the cortex.	EOAD	[102]
	Met233Val	Cerebral cortex, spinal cord	Cerebral cortex	Leptomeningeal, cerebral, and cerebellar vessels	Lewy bodies in the substantia nigra and cortex.	EOAD Lewy bodies	[60] [106]
	Val272Ala	Cortex, thalamus, hypothalamus and mesencephalon, substania nigra	Cortex	NA	Lewy bodies in the cortex and substantia nigra.	Subcortical dementia and parkinsonism	[104]
	Ala396Thr	neocortex and basal ganglia	Neocortex, allocortex, substantia nigra, and locus coeruleus	Diffuse plaques and vessels in the entire brain	Lewy bodies in cerebral cortex, caudate nucleus, putamen, hippocampus, and substantia nigra locus coeruleus. Alpha synuclein and Tau may co-aggregate.	AD and DLB	[103]
	Ala431Glu	Several brain areas	Several brain areas	NA	Lewy bodies in amygdala, cingulate gyrus, and neocortex in one patient.	AD and DLB	[107]
	Thr440fs	cortex, hippocampus, Substantia nigra, pons, medulla	Hippocampus, pons,	Vessels in cerebrum and cortex	Alpha-synuclein-positive Lewy bodies in several brain areas and cotton wool plaques.	AD and DLB	[57]

## 4. Discussion

AD and other neurodegenerative diseases (such as FTD, PD or DLB) may share common hallmarks. In terms of symptoms, cognitive and movement impairments are the main clinical feature of most diseases. Neurodegenerative diseases are considered proteiopathies, associated with abnormally folded proteins which could aggregate and accumulate in the brain. These protein deposits lead to neurodegeneration through apoptosis or autophagy. Different protein deposits have been described, for example, amyloid beta aggregates and NFTs in AD, Lewy bodies in DLB or PD, tauopathies or Pick’s bodies in FTD and progressive primary aphasia and TDP-43 proteinopathies in FTD or ALS. However, there may be overlap between these aggregates, for example, Pick’s bodies, syunucleopathies and Lewy bodies may occur in AD patients [108,109,110]. These findings suggest that neurodegenerative diseases may share common pathways [111]. Genetic overlap between neurodegenerative diseases can also be possible. For example, hexabase repeat expansion in c9orf72 promoter could impact both FTD and ALS. Repeat expansions in spinocerebellar-ataxia-related genes (SCA2 or SCA3) may be related to PD. LRRK2 is a PD-causing gene, but LRRK2-associated neuropathology in patients may be diverse (such as Lewy bodies or alpha synuclein aggregates). The MAPT gene was verified as causative factor for FTD, but its role in AD or PD could also be discussed [110]. These findings suggest that PSEN1 may also impact disease phenotypes other than EOAD. 

PSEN1 was verified as a multi-functional protein, which may impact several mechanisms besides APP processing. It may play critical role in Notch signaling, the beta-catenin-related calcium balance, cell trafficking or cell survival [3]. In addition to APP and Notch, the gamma secretase complex has multiple substrates, including cadherins, LRP, CD43, CD44 and tyrosinase proteins [112]. PSEN1 mutations were associated with diverse phenotypes. Several PSEN1 mutation carriers did not present classical EOAD phenotypes, but other neurodegenerative diseases, such as FTD (such as Leu113Pro and Thr122Ala), DLB (such as Glu184Asp and Ala275Ser), PD, Parkinsonism, spastic paraparesis (such as Arg41Ser and Glu120Lys). These findings suggest that PSEN1 may contribute to their pathogenesis. These cases did not fulfill the diagnostic criteria for AD (CERAD and NIA-Reagan criteria). These findings suggested that PSEN1 may be a causative factor for non-AD-type neurodegeneration or act as risk modifier by interacting with FTD or PD risk genes. The interaction between Tau and PSEN1 [44,45], PSEN1-PRKN-PINK1 [53] and PSEN1-alpha-synuclein [63] has been verified. These interactions may be responsible for the atypical disease phenotypes or non-AD phenotypes.

Another potential proof of gene interaction could be the possible phenotypic diversity of a single PSEN1 mutation. For example, Met146Val was discovered in a family with FTD [39], but it also appeared in AD cases [113,114]. Met146Leu was also reported in multiple families; one of them had Pick’s disease [89], while the others developed EOAD [112,113]. Glu184Asp appeared in a family with DLB [37], while the other families had EOAD phenotypes [115,116]. With the development of next-generation sequencing or whole genome/exome sequencing techniques, the potential disease-modifying factors and the interaction between PSEN1 and other disease factors may be discovered more easily [117,118,119]. Interestingly, PSEN1 Pro242fs and Asp333Gly could impact acne inversa and DCM, respectively. Notch signaling impairments were associated with disease phenotypes in both cases, but these mutations may also be related with either abnormal inflammatory reactions or a calcium balance impairment [66,67,68,69,70].

Taken together, PSEN1 was verified as a complex protein, which may be related to diverse disease phenotypes such as FTD, DLB or motor impairments. PSEN1 may also impact non-neurodegenerative diseases. Further studies are needed to determine whether PSEN1 is truly a causative factor for these diseases or if it is only a contributing factor. PSEN1 was verified to interact with other disease-related genes and proteins through biomarkers, such as amyloid-beta species, oligomers, alpha synuclein or Tau [120,121,122,123]. Next-generation sequencing analyses may be needed to discover the possible genetic factors that could play a role in the alternative disease phenotype associated with PSEN1 mutation. 

## Figures and Tables

**Figure 1 ijms-24-08417-f001:**
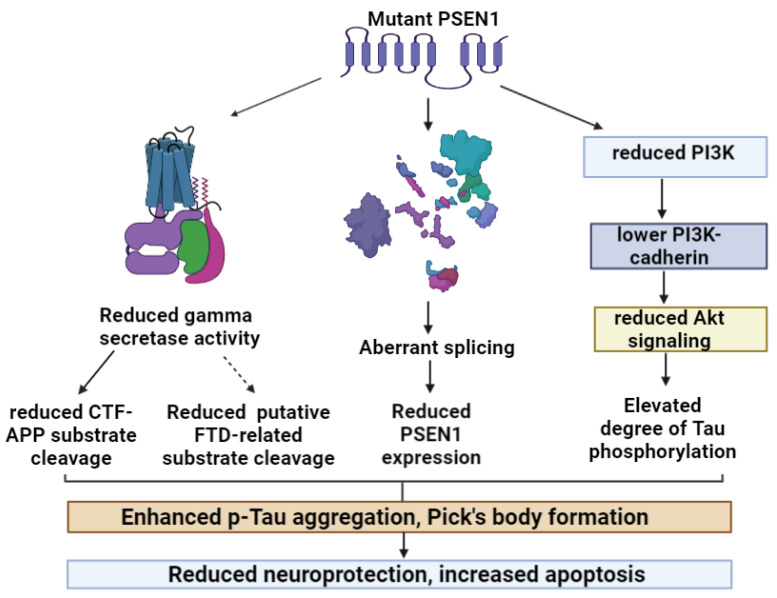
Possible pathways of PSEN1 mutations, leading to FTD.

**Figure 2 ijms-24-08417-f002:**
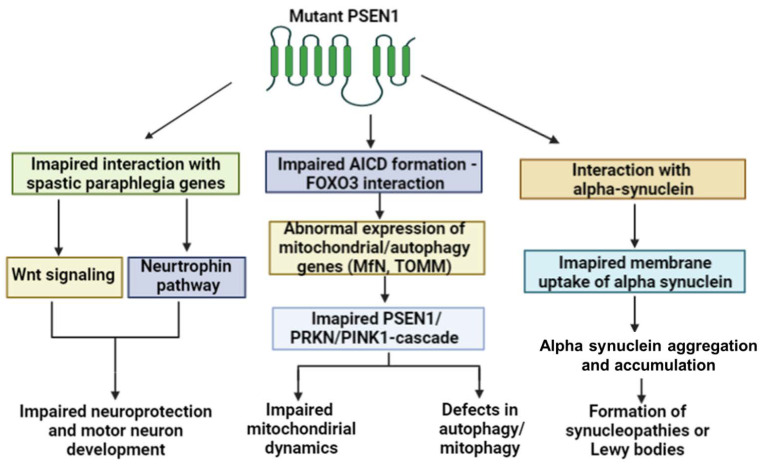
Possible association between PSEN1 mutations and motor diseases.

**Figure 3 ijms-24-08417-f003:**
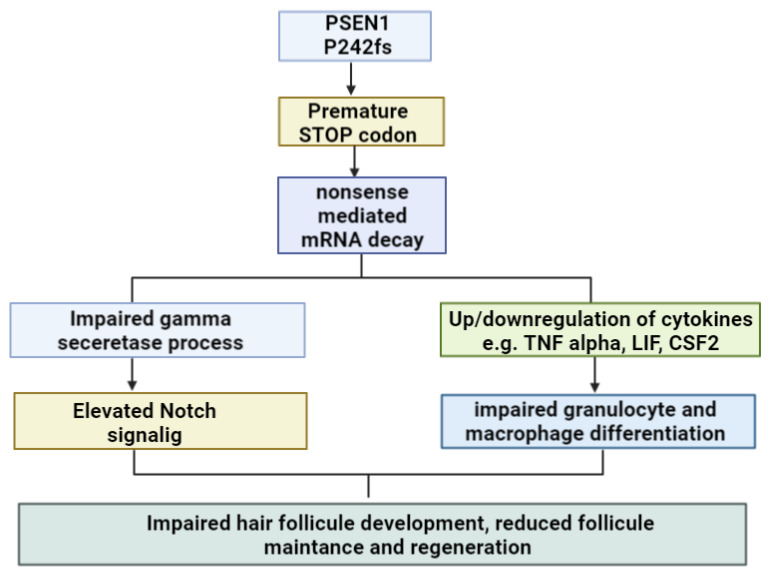
Possible impact of PSEN1 Pro242fs in acne inversa [66,67,68,69,70].

**Figure 4 ijms-24-08417-f004:**
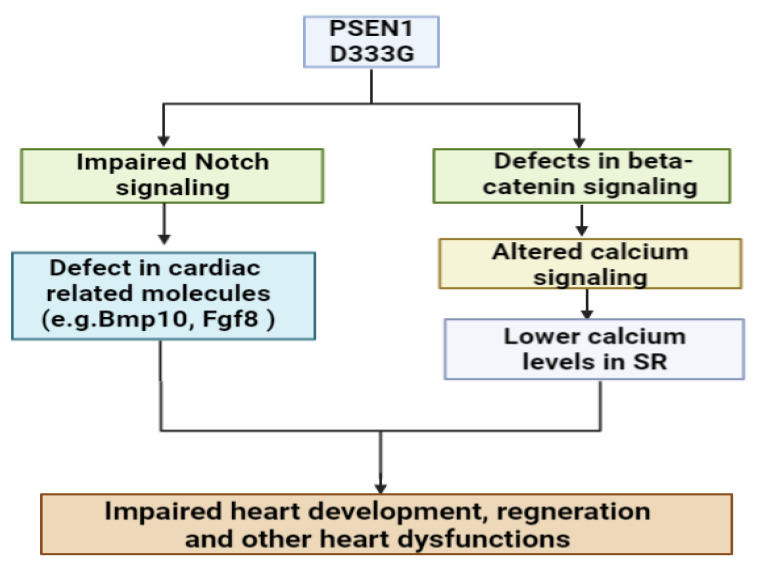
Possible impact of PSEN1 Asp333Gly in DCM [71,72,73,74,75,76,77,78,79].

**Table 1 ijms-24-08417-t001:** Examples of PSEN1 mutations observed in at least one case with a form of neurodegeneration other than AD [Alzforum].

Disease	Mutation
FTD or progressive non-fluent aphasia or Pick’s disease	Gln15His; Pro88Arg; Leu113Pro; Thr122Ala; Ser132Ala; Ala137Thr; Met146Val; Met146Leu; Gly183Val; Leu166Pro; Leu226Phe; Met233Leu; Arg278Ile; Pro264Leu Pro303Leu; Arg352dup; Val412Ile
ALS	Leu166Pro; Trp203Cys; Ile249Leu
PD	Arg41Ser; Leu85Pro; Glu120Lys; Tyr156Cys; Ser170Pro; Gly217Asp; Pro264Leu; Val272Ala; Tyr288His; Tyr389His; Val391Gly; Gly417Ala; Ala434Thr
Spastic paraparesis	Ile83_Met84del; Met84Val; Leu85Pro; Glu120Lys; Tyr154Asn; Tyr156_Arg157insIle_Tyr; Leu166Pro; Gln223Arg; Phe237Ile; Val261Leu; Val261Phe; Pro264Leu; Gly266Ser; Arg278Thr; Arg278Lys; Arg278Ser; Glu280Gly;Pro284Ser; Pro284Leu; Tyr288His; Ser290Cys; Thr291Ala; Thr291Pro; Leu381Val; Phe388Ser; Gly417Ser; Leu424Arg; Pro436Gln
Ataxia	Pro117Ala; Thr147Pro; Met233Val
DLB	Ser132Ala; Ala275Ser; Thr440del
Non-neurodegenerative	Pro242fs; Asp333Gly

## Data Availability

Not applicable.

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
