# Peer review of "Presenilin-1 (PSEN1) Mutations: Clinical Phenotypes beyond Alzheimer’s Disease"

_ijms, 2023, doi:10.3390/ijms24098417_

Round 1

Reviewer 1 Report

This manuscript presents detailed information regarding the connections between PSEN1 mutations and some motor diseases. Overall, the whole manuscript is ok. But a few things need to be improved before accepting.

1) Too much information has been inserted in the abstract. You may want to put a general description in this part, otherwise, it's unreadable.

2) Figure1: the down arrows included in the boxes of PI3K, cadherin, and Akt are very confusing to me.

3) you can discuss the potential treatments by using the mutants as a starting point. 

Author Response

This manuscript presents detailed information regarding the connections between PSEN1 mutations and some motor diseases. Overall, the whole manuscript is ok. But a few things need to be improved before accepting.

  • Too much information has been inserted in the abstract. You may want to put a general description in this part, otherwise, it's unreadable.

Thank you for the suggestion, abstract has been reduced.

  • Figure1: the down arrows included in the boxes of PI3K, cadherin, and Akt are very confusing to me.

Thank you, the Figure 1 has been fixed for better understanding.

  • you can discuss the potential treatments by using the mutants as a starting point. 

Thank you for the great suggestion, we added a paragraph in the introduction on the promises of gene therapies and a few examples of gene therapy approaches in PSEN1 mutations (cell models).

“Discovery of PSEN1 mutations in AD and other diseases may be promising targets in gene therapy. Currently, there is no effective therapy for majority of neurodegenerative diseases, including genetic diseases. However, gene therapies may be promising approaches in the treatment, since they were successfully used against spinal muscular atrophy (SMA). Functional replacement of mutant gene was associated in children with SMA was associated with improved motor functions [121]. Similar studies are going on to treat genetic AD by “fixing” the mutant PSEN1. However, these studies were only tested in in vitro cell models. Human iPSC lines from an AD patient with Leu150Pro and Ala79Val were also targeted to CRISPR-Cas9 gene correction, and both approaches were successful. These studies may be useful in future studies of gene therapies. [122, 123]. CRISPR-Cas9-based treatment was used against PSEN1 Met146Leu in fibroblasts. Guide RNAs (gRNAs) were designed against mutant PSEN1. Knocking out the mutant PSEN1 was associated with reduced Ab42/40 ratio [124]. Gene silencing may also be promising in case of PSEN1 mutations. HeLa cells and fibroblasts with PSEN1 Leu392Val mutations were targeted to antisense siRNAs, which were designed to the mutation (with one mismatch to the no-mal gene). The gene silencing reduced in toxic Ab42 production [125]. Identifying the dis-ease risk (including PSEN1 mutation carrier) individual at preclinical stage should be essential in for future therapies, since gene therapies may be effective before clinical symptoms appear [124]"

Reviewer 2 Report

In the abstract, the use of acronyms and their explanation in brackets should be standardised). Sometimes the initials are described and sometimes not, as in the case of EOAD, AD, ALS or MAPT. Which, by the way, are then described in the article.

Paragraphs are too long, some are more than 40 lines long, and do not allow the reader to stop and reflect on what they have just read. Even the summary is too long.

The sections of the review could be improved. If there is a section headed PSEN1 potential impact on other forms of neurodegenerative disease, the reader would expect a section on PSEN1 in neurodegenerative disease before that. 

English language is fine

Author Response

In the abstract, the use of acronyms and their explanation in brackets should be standardized). Sometimes the initials are described and sometimes not, as in the case of EOAD, AD, ALS, or MAPT. Which, by the way, are then described in the article.

Thank you, Reviewer 1 also suggested reducing the size of the abstract. This part from the abstract has been removed.

Paragraphs are too long, some are more than 40 lines long, and do not allow the reader to stop and reflect on what they have just read. Even the summary is too long.

Thank you, the issue has been fixed, paragraphs have been fragmented for better understanding. The size of the summary has also been reduced.

The sections of the review could be improved. If there is a section headed PSEN1 potential impact on other forms of neurodegenerative disease, the reader would expect a section on PSEN1 in neurodegenerative disease before that. 

Thank you, we rearranged the sections of the manuscript. Currently, the chapter on “Atypical inclusions in the brain of patients with PSEN1 mutations” is Chapter 3, and the long chapter on “PSEN1 potential impact in other forms of neurodegenerative disease’ is Chapter 2